# Methods of Esthetic Assessment after Adjuvant Whole-Breast Radiotherapy in Breast Cancer Patients: Evaluation of the BCCT.core Software and Patients’ and Physicians’ Assessment from the Randomized IMRT-MC2 Trial

**DOI:** 10.3390/cancers14123010

**Published:** 2022-06-18

**Authors:** Tobias Forster, Clara Köhler, Melissa Dorn, Matthias Felix Häfner, Nathalie Arians, Laila König, Semi Ben Harrabi, Ingmar Schlampp, Eva Meixner, Vanessa Heinrich, Nicola Weidner, Michael Golatta, André Hennigs, Jörg Heil, Holger Hof, David Krug, Jürgen Debus, Juliane Hörner-Rieber

**Affiliations:** 1Department of Radiation Oncology, Heidelberg University Hospital, 69120 Heidelberg, Germany; tobias.forster@med.uni-heidelberg.de (T.F.); koehler@stud.uni-heidelberg.de (C.K.); melissadorn97@gmail.com (M.D.); matthias.haefner@med.uni-heidelberg.de (M.F.H.); nathalie.arians@med.uni-heidelberg.de (N.A.); laila.koenig@med.uni-heidelberg.de (L.K.); semi.harrabi@med.uni-heidelberg.de (S.B.H.); ingmar.schlampp@med.uni-heidelberg.de (I.S.); eva.meixner@med.uni-heidelberg.de (E.M.); david.krug@uksh.de (D.K.); juergen.debus@med.uni-heidelberg.de (J.D.); 2Heidelberg Institute of Radiation Oncology (HIRO), 69120 Heidelberg, Germany; 3National Center for Tumor Diseases (NCT), Heidelberg University Hospital, 69120 Heidelberg, Germany; 4Department of Radiation Oncology, Eberhard-Karls-University Tuebingen, 72070 Tuebingen, Germany; vanessa.heinrich@med.uni-tuebingen.de (V.H.); n.weidner@med.uni-tuebingen.de (N.W.); 5Department of Gynecology and Obstetrics, University of Heidelberg, 69115 Heidelberg, Germany; michael.golatta@med.uni-heidelberg.de (M.G.); andre.hennigs@med.uni-heidelberg.de (A.H.); joerg.heil@med.uni-heidelberg.de (J.H.); 6Strahlentherapie Rhein-Pfalz, 67433 Neustadt, Germany; hof@strahlentherapie-neustadt.de; 7Department of Radiation Oncology, University Hospital Schleswig Holstein, 24105 Kiel, Germany; 8Clinical Cooperation Unit Radiation Oncology, German Cancer Research Center (DKFZ), 69120 Heidelberg, Germany; 9Heidelberg Ion-Beam Therapy Center (HIT), Department of Radiation Oncology, Heidelberg University Hospital, 69120 Heidelberg, Germany; 10German Cancer Consortium (DKTK), Partner Site Heidelberg, Heidelberg University Hospital, 69120 Heidelberg, Germany

**Keywords:** breast cancer, adjuvant whole-breast radiotherapy, esthetic assessment, BCCT.core software, cosmesis

## Abstract

**Simple Summary:**

To validate the BCCT.core software, the present analysis compares the esthetics assessment by the software in relation to patients’ and physicians’ rating in breast cancer patients after surgery and adjuvant radiotherapy. Agreement rates of the different assessments and their correlation with breast asymmetry indices were evaluated. The assessments of the software and the physicians were significantly correlated with all asymmetry indices, while for patients’ self-assessment, this general correlation was first seen after 2 years. Only a slight agreement between the BCCT.core software and the physicians’ or patients’ assessment was seen, while a moderate and substantial agreement was detected between the physicians’ and the patients’ assessments. The BCCT.core software is a reliable tool to measure asymmetries, but may not sufficiently evaluate the esthetic outcome as perceived by patients. It may be more appropriate for a long-term follow-up, when symmetry seems to increase in importance.

**Abstract:**

The present analysis compares the esthetics assessment by the BCCT.core software in relation to patients’ and physicians’ ratings, based on the IMRT-MC2 trial. Within this trial, breast cancer patients received breast-conserving surgery (BCS) and adjuvant radiotherapy. At the baseline, 6 weeks, and 2 years after radiotherapy, photos of the breasts were assessed by the software and patients’ and physicians’ assessments were performed. Agreement rates of the assessments and their correlation with breast asymmetry indices were evaluated. The assessments of the software and the physicians were significantly correlated with asymmetry indices. Before and 6 weeks after radiotherapy, the patients’ self-assessment was only correlated with the lower breast contour (LBC) and upward nipple retraction (UNR), while after 2 years, there was also a correlation with other indices. Only a slight agreement between the BCCT.core software and the physicians’ or patients’ assessment was seen, while a moderate and substantial agreement was detected between the physicians’ and the patients’ assessment after 6 weeks and 2 years, respectively. The BCCT.core software is a reliable tool to measure asymmetries, but may not sufficiently evaluate the esthetic outcome as perceived by patients. It may be more appropriate for a long-term follow-up, when symmetry appears to increase in importance.

## 1. Introduction

Due to the early detection of breast cancer, increasing rates of breast conservation, and improvements in long-term survival, the esthetic outcome of breast-conserving therapy is coming more and more into focus for both caregivers and caretakers. In terms of radiation oncology, the use of boost irradiation is a major determinant of esthetic outcome [1,2,3]. Thus, for the evaluation of new radiotherapy techniques, the esthetic outcome is undoubtedly a highly important outcome parameter. Different methods are reported in the literature to measure the esthetic outcome: subjective approaches, such as patients’ self-assessment and third-party assessment performed by a single physician or a panel of physicians, and objective techniques, primarily measuring symmetry [4,5,6]. Subjective panel evaluation and self-assessment continue to be the most commonly used methods for evaluating esthetic outcome [7]. To improve the reproducibility and comparability of subjective esthetics assessments, the standardized four-point Harvard Scale was introduced by Harris et al., and broadly used by the research community [8]. Nevertheless, in breast cancer therapy, the reproducibility of subjective assessments of esthetic outcome still remains limited [9,10]. Furthermore, a subjective evaluation performed by a panel of experts is an expensive, difficult, and time-consuming procedure [10]. Therefore, a reliable, highly reproducible, and simple method for esthetic assessment is urgently needed for future research approaches. The BCCT.core (Breast Cancer Conservative Treatment. Cosmetic Result) software is a cost-free, semi-automated, and easy-to-use tool, which provides a highly reliable and reproducible assessment of esthetic outcome [11,12,13,14]. In previous studies, the validity of the BCCT.core software was tested by comparing the results of the program to a subjective panel assessment [11,12,15]. Furthermore, the agreement of the software results with the patient perspective was tested in comparison to the BCTOS (Breast Conservative Treatment Outcome Scale) Aesthetic Status [13]. However, the number of cases tested in these trials were rather small, consisting of 30 to 128 patients. In the present study, the esthetic outcome data of the prospective, two-armed, randomized phase III IMRT-MC2 trial were used for an in-depth validation of the BCCT.core software with a larger number of cases. For this purpose, the software results were analyzed in comparison to physicians’ assessments and patients’ self-assessment of the esthetic outcome.

## 2. Results

The IMRT-MC2 trial randomized 502 patients to the control arm (3-D-CRT-seqB) and the experimental arm (IMRT-SIB) in a 1:1 ratio. For a total of 472 patients, 433 patients, and 378 patients, a complete assessment of esthetic outcome was available at the baseline, at 6 weeks, and 2 years after radiotherapy, respectively. These patients were eligible for the present analysis. For the three time points, the characteristics of patients, tumors, and treatments are summarized in Table 1. At the baseline, the median age of the participants was 55 years. Most patients presented with T1 stage disease (74%), and were staged pN0 (80%). In most cases, the tumor was located in the upper outer quadrant of the breast (61%).

### 2.1. Correlation between Breast Asymmetry Indices and Overall Esthetic Scores

The correlations between different breast asymmetry indices and overall esthetic outcomes of the BCCT.core software, as well as between breast asymmetry indices and patients’ or physicians’ assessment scores, at the baseline, 6 weeks, and two years after radiotherapy are shown in Table 2.

For the overall esthetic score of the BCCT.core software, a significant correlation was seen with all asymmetry indices at all time points. The highest Pearson correlation score was detected for the lower breast contour (LBC), which indicates the difference between the level of the inferior breast contours (Pearson coefficient: 0.669, 0.554, and 0.688 for the three time points, respectively; *p* < 0.001).

At the baseline, 6 weeks, and two years after radiotherapy, the overall physicians’ assessment of esthetic outcome also significantly correlated with all breast asymmetry indices. For physicians’ assessments of esthetics, the Pearson correlation coefficients were lower than the correlation coefficients seen for the cosmetic score of the BCCT.core software, indicating a stronger correlation of asymmetry indices with the score of the BCCT.core software.

Only for LBC and upward nipple retraction (UNR) was a significant correlation with the patients’ self-assessment of esthetics seen at all time points. However, two years after radiotherapy, all asymmetry indices were significantly correlated with the patients’ self-assessment.

When the four-point scale of esthetics was dichotomized into a two-point scale, no correlation between breast asymmetry indices and patients’ self-assessment of esthetics was seen for the baseline and 6 weeks after radiotherapy time points. Otherwise, no differences were seen between the analysis of the four-point and the two-point scale (Table 3).

### 2.2. Agreement of BCCT.Core Software Results with Patients’ and Physicians’ Assessment Scores

The agreement of the BCCT.core software results for esthetics with patients’ or physicians’ assessment scores, as well as the agreement of patients’ assessment scores with physicians’ assessment scores, are depicted in Table 4 for the baseline, 6 weeks, and two years after radiotherapy.

At the baseline, there was only a slight agreement between the BCCT.core software results and patients’ self-assessment scores (weighted Kappa (wk) = 0.109; *p* = 0.003), as well as between patients’ and physicians’ assessment scores (wk = 0.100; *p* < 0.001). In the time points of 6 weeks and 2 years after radiotherapy, only a slight agreement was seen for the BCCT.core esthetic results and physicians’ assessment scores (wk = 0.084; *p* = 0.001, wk = 0.138; *p* < 0.001, respectively). For the correlation of BCCT.core results and patients’ self-assessment two years after therapy, there was also only slight agreement (wk = 0.111; *p* = 0.002). On the other hand, moderate and even substantial agreement was seen for patients’ and physicians’ assessment scores both 6 weeks (wk = 0.572; *p* < 0.001) and two years (wk = 0.625; *p* < 0.001) after radiotherapy, respectively.

In general, an increase in agreement was seen over time: With increasing time since therapy, the agreement also increased, with the highest agreement rates presented after two years, and the lowest values at the baseline. After dichotomizing the four-point scale of esthetics, there was no substantial change in the results of the analysis. However, the level of agreement was generally lower (Table 5).

## 3. Discussion

As the overall esthetic score of the BCCT.core software is primarily based on asymmetry measurements, a strong correlation with breast asymmetry indices was detected in the present analysis at all time points. This effect was expected, and was strongest for LBC, which is in line with results of Yu et al., also reporting the highest Pearson correlation score for this asymmetry index [16]. Physicians’ assessment of esthetics also significantly correlated with breast asymmetry indices. Although the effect was not as strong as for the BCCT.core software, it demonstrates that physicians’ assessments also appear to be predominantly influenced by asymmetry. This is in line with results of Lyngholm et al., who analyzed late morbidity, esthetic outcome, and body image of 214 patients from the Danish Breast Cancer Cooperative Group after breast-conserving therapy. This study demonstrated that breast asymmetry measured by BRA was the only factor that correlated with physicians’ assessments of esthetic outcome [17].

For patients’ self-assessment, a significant correlation with asymmetry indices at the baseline and 6 weeks after radiotherapy mark was only seen for LBC and UNR in the present study. Other asymmetry scores, such as the pBRA, significantly correlated with patients’ assessments only after two years. Assumingly, for patients immediately after surgery and radiotherapy factors, other than asymmetry, such as scars, fibrosis, hyperpigmentation, or even pain, may have a stronger influence on esthetic self-assessment. Then, as time passes, the symmetry of the breast may rise again in importance. This effect may also explain the stabilization of the esthetic outcome 3 years after therapy, as it is found in most previous studies [18,19,20,21,22]. As an uneven lower breast contour and nipple position, measured with LBC and UNR, are the most eye-catching parameters of asymmetry, it may be reasonable that a correlation of these factors and patients’ esthetics assessments was detected earlier after treatment. Only a weak correlation of patients’ esthetics rating with asymmetry indices, as seen in the present work, was also observed by Yu et al.; at a median follow-up of two years after breast-conserving therapy, the authors evaluated the esthetic outcome rated by the 51 patients themselves, and did not find any significant correlation with breast asymmetry measurements [16]. Some authors argue that assessment of esthetics with a four-point scale is less appropriate for patients, and that these global rating categories, when used by patients themselves, may only reflect perceived differences between treated and untreated breasts [23,24]. In an analysis of Patterson et al., 94% of patients rated the esthetic outcome after breast-conserving therapy as good or excellent, although 50% of patients reported relevant differences between the treated and the untreated breast [25]. However, rating by physicians may be considered to be fairly subjective as well.

The present analysis indicates a stronger agreement for physicians’ assessments of esthetic outcome (moderate and substantial agreement 6 weeks and 2 years after therapy, respectively) than for the assessment with the BCCT.core software (only slight agreement). This is in line with similar agreement rates detected in previous studies; in an analysis of Heil et al.’s study, which included 128 patients, a slight to fair agreement was seen between the BCCT.core results and patients’ assessments of esthetics, tested with the BCTOS Aesthetic Status questionnaire [13]. A study of Yu et al., including 51 patients, also demonstrated only a slight agreement between BCCT.core software results and patients’ self-assessment 2 years after whole-breast radiotherapy [16].

In the current analysis, the observed effect of a stronger agreement with physicians’ assessments than with the BCCT.core results may be due to a more holistic and less asymmetry-based evaluation of the breast by the physicians, taking into account other relevant factors, such as scars and pigmentation. Representative examples of agreement and disagreement of the software results with patients’ and physicians’ assessment scores are depicted in Figure 1. However, two years after radiotherapy, an increase in agreement rates, as compared to patients’ self-assessment, was detected for both the physicians’ assessment and the assessment by the software. This observed increase in agreement levels for the software results with patients’ self-assessment may be due to a rising importance of symmetry for the patients with time. Therefore, the BCCT.core software appears to be a better measurement method for long-term follow-up of the esthetic outcome, rather than evaluating the esthetic results immediately after therapy.

When comparing our findings to the results of the previous studies of Haloua et al., Cardoso et al. and Heil et al., we could not confirm similar high rates of agreement of physicians’ assessment and BCCT.core software results (reported Kappa: 0.27 to 0.64) [11,12,15,26]. This may partly be explained by the fact that, in those studies, the physicians’ rating was performed by a panel, rather than by a single physician, as was the case in the present prospective phase III trial. For the current analysis, physicians’ assessment was performed by live observation and palpation, as recommended by previous authors [27], to detect outcome variables such as edema, fibrosis, and telangiectasia more easily. In this setting, a panel evaluation was not feasible. Although intra-rater agreement in third-party assessment of the esthetic outcome is described to be substantial in the literature [28], a suboptimal intra-rater agreement may have weakened the present analysis. Unfortunately, intra-rater agreement was not tested in the present work. Moreover, the rating physician was one of the treating radiation oncologists. This may have influenced the physicians’ assessment towards a more positive esthetic outcome.

A strength of the present study, in addition to the large number of patients assessed, is the use and comparison of all three methods of esthetics assessment (subjective, objective, and self-assessment) as recommended in the literature [7]. In particular, the results of patients’ self-assessment provide a highly relevant reference for the present analysis, as well as for future studies. Furthermore, in the current analysis, the application of the standardized four-point scale of Harris et al., in combination with photographic examples [8,18], as recommended by Vrieling et al. [24], makes it easy to compare our results to those of other authors.

According to our analysis, the BCCT.core software is an excellent tool for reliably measuring asymmetry, which appears to be a strong influencing factor for physicians’ assessment of esthetics. However, in the first weeks after breast-conserving therapy, patients’ self-assessment appears to depend less on asymmetry indices. Patients’ self-assessment is certainly the most relevant method to measure the esthetic outcome, even if this is not objective. Nevertheless, the reliability of patients’ assessment and its use for validation purposes is controversially discussed in the literature, as subjective assessments may not necessarily agree with the objective scores, and have very low reproducibility values [11,13,24,29]. In the literature, the application of both a qualitative and a quantitative method of measuring the esthetic outcome is recommended, especially when skin changes such as scars are relevant, since a single assessment that covers all of this complexity will probably never exist [18,24].

## 4. Patients and Methods

### 4.1. Description of Analyzed Patients

The present investigation was based on the esthetic outcome of participants of the IMRT-MC2 prospective, two-armed, multi-center, randomized phase III trial at the baseline of, as well as 6 weeks and 2 years after, whole-breast irradiation (WBI). The multicenter IMRT-MC2 trial recruited from March 2011 until August 2015 at the University Hospital Tübingen and the University Hospital Heidelberg. The University of Heidelberg ethics committee (S-041/2009) and the Federal Office of Radiation Protection (BfS) (Z5-22461/2-2009-18) (ClinicalTrial.gov Protocol ID is NCT01322854) approved the protocol. The German Aerospace Center (DLR)/Federal Ministry of Education and Research (BMBF) of Germany (01ZP0504) funded the IMRT-MC2 trial. All patients received breast-conserving surgery, followed by adjuvant WBI with boost irradiation to the former tumor bed. Adjuvant WBI was delivered either with IMRT (helical tomotherapy or step-and-shoot technique) to a total dose of 50.4 Gy in 1.8 Gy single fractions with a simultaneous integrated boost (SIB) to the tumor bed to a total dose of 64.4 Gy in 28 fractions of 2.3 Gy (IMRT-SIB), or with 3D-conformal radiotherapy (3D-CRT) to the whole breast to a total dose of 50.4 Gy and 1.8 Gy per fraction in 28 fractions, followed by a sequential boost (seqB) to a total dose of 66.4 Gy at 2 Gy dose per fraction in 8 fractions (3D-CRT-seqB) [30,31]. Detailed information about target volume delineation, dose prescription, dose constraints, and treatment planning have been published previously [30,31].

All participants of the IMRT-MC2 trial with a complete assessment of esthetic outcome were included in the present analysis. All patients of the present analysis gave their written informed consent to participate in the study, and were characterized by the following criteria: all patients had to have an indication for adjuvant WBI with boost irradiation to the former tumor bed. Patients needed to be aged ≥ 18 years and < 70 years, or aged ≥ 70 years with one of the following risk factors: multifocal disease, tumor stage ≥ T2, extensive intraductal component, lymphangiosis and resection margins ≤ 3 mm, Karnofsky Performance Score > 70%, no metastatic disease (M0), no previous radiotherapy of the same breast or thorax, no other malignancies in the previous 5 years, no pregnancy [30].

### 4.2. Software Analysis of Esthetic Outcome Using BCCT.Core Software

Each patient of the present analysis received a standardized photographic documentation of the breasts shortly before radiotherapy (baseline), as well as 6 weeks and 2 years after irradiation. Using the following same procedure for every woman, frontal photographs of both breasts excluding the face were taken: standing position with hanging arms, equal standardized exposure to light, equal distance of 2.5 m from camera to patient, equal background in blue color. The sternal notch, as well as a point 25 cm below, were marked with red dots to allow for the correction of the magnification of the photographs. For each patient and for every single time point, these photographs were analyzed using the BCCT.core software (Breast Research Group, Porto, Portugal) to calculate breast asymmetry indices and an overall score of the esthetic result. This software was developed to summarize all objective symmetry measurements ever described in one single tool [5,6,7,9,15,18,32]. After importing a standardized photograph into the software, predetermined points are designated by the user, and the breast contour is semi-automatically delineated (Figure 2). In a following step, the BCCT.core software calculates different asymmetry indices, including breast volume, skin color, and scars. Finally, an algorithm combines these indices in an overall esthetic result, displayed on a four-point scale (1 = excellent, 2 = good, 3 = fair, 4 = poor). All measurements and calculations were performed by the same investigator (TF) who was blinded regarding timepoint and randomization.

### 4.3. Patients’ and Physicians’ Assessment of Esthetic Outcome

Shortly before radiotherapy (baseline), as well as 6 weeks and 2 years after irradiation, the esthetic outcome was assessed by both the treating physician and the patient using the Radiation Therapy Oncology Group/Harvard Scale, comparing the treated breast to the untreated one. The esthetic outcome was scored according to a four-point scale: excellent (no visible treatment sequelae at first sight), good (minimal changes), fair (the treated breast is different but not seriously distorted), or poor (the treated breast is seriously distorted) [8]. The rating physicians were all radiation oncologists, and had at least 2 years’ experience in the field.

### 4.4. Statistical Methods

For the agreement analyses of patients’ and physicians’ subjective scoring and the objective software scoring of esthetic outcome, absolute agreement rates (a), Kappa (k), and weighted Kappa (wk) statistics were used [33,34]. The agreement analysis was performed separately for the 3 different times of assessment: baseline (before radiotherapy), 6 weeks, and 2 years after radiotherapy. To interpret the Kappa and weighted Kappa coefficients, we used the definition recommended by Seigel et al.: ≤0 indicates poor agreement, 0.01–0.20 slight agreement, 0.21–0.40 fair agreement, 0.41–0.60 moderate agreement, 0.61–0.80 substantial agreement, 0.81–0.99 almost perfect agreement, and 1.00 perfect agreement [35]. Using absolute agreement rates (a), Kappa (k), and weighted Kappa (wk), we tested agreement of the esthetic scores (four-point scale) of the BCCT.core software and the patients’ subjective scoring, of the BCCT.core software and the physicians’ subjective scoring, and of the patients’ and the physicians’ subjective scoring. In a second step, the four-point scale of the patients’, physicians’, and software assessment was dichotomized into “excellent/good” and “fair/poor” esthetic outcomes, and the statistical analysis was repeated for this two-point scale.

For the correlation analyses of different breast asymmetry indices and subjective patients’, subjective physicians’, and objective software scores for overall esthetic outcome (four-point scale), a Pearson correlation coefficient was used [36]. The correlation analysis was performed separately for the three different times of assessment, as mentioned above. Again, the four-point scale of the esthetic outcome was dichotomized into “excellent/good” and “fair/poor”, and the correlation analysis was repeated for the two-point scale.

Data analyses were performed with the IBM Statistical Package for Social Sciences software, version 19 (SPSS Inc., Chicago, IL), to calculate Kappa, weighted Kappa, the Pearson coefficient, and *p*-values. A two-sided significance test was used to compute the statistical significance. The level of statistical significance was set at *p* ≤ 0.05.

## 5. Conclusions

The presented data demonstrate that an assessment by the BCCT.core software alone may not be sufficient to evaluate the esthetic outcome in the way that it is perceived by the patients. The BCCT.core software is a good and reliable tool to measure objective asymmetries, but it should be complemented by physicians’ assessment and patients’ self-assessment to take the subjective perception of esthetics into account. The BCCT.core software may be more appropriate for measuring the esthetic outcome in a long-term follow-up, when symmetry appears to become more important for patients.

## Figures and Tables

**Figure 1 cancers-14-03010-f001:**
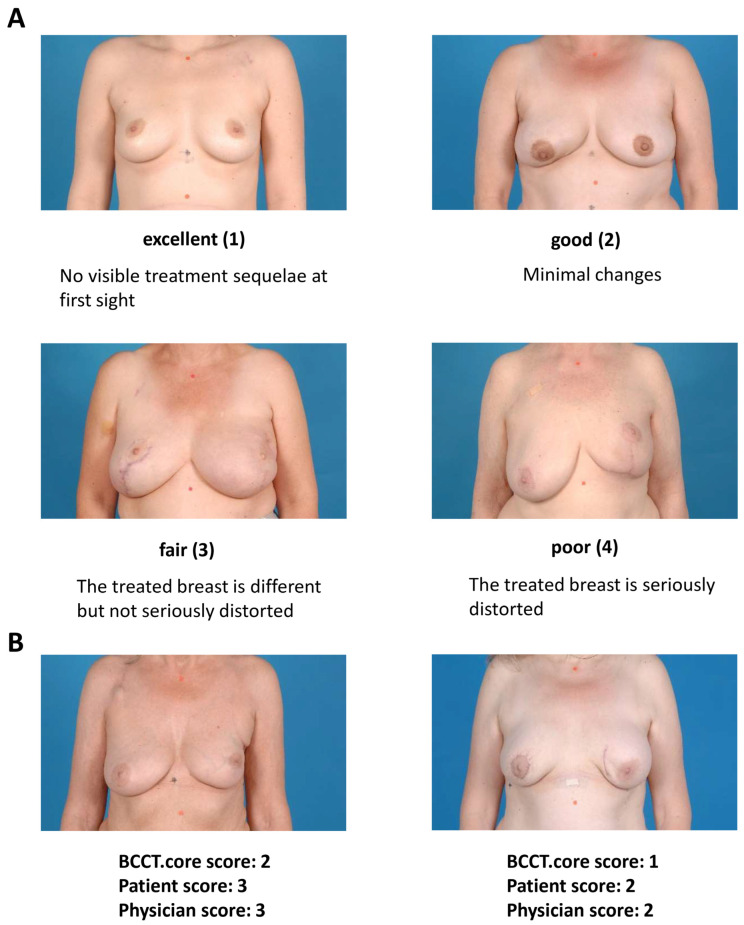
Esthetic results assessed with BCCT.core. Representative examples of excellent outcome, good outcome, fair outcome, and poor outcome, with agreement of the BCCT.core software results with patients’ and physicians’ assessment scores shown together with the definition of the Harvard Scale (**A**), as well as representative examples of disagreement (**B**).

**Figure 2 cancers-14-03010-f002:**
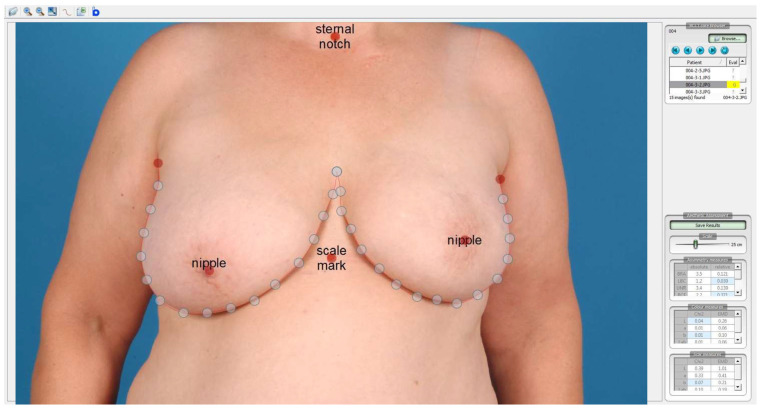
Screenshot of BCCT.core software. The sternal notch, a point 25 cm below, and both nipples are marked. After the user has adjusted the red dots to the most medial and lateral point of the breast outline, the BCCT.core software is able to auto adjust the breast outline and to calculate several asymmetry indices, as well as an overall score (four-point scale: excellent–good–fair–poor).

**Table 1 cancers-14-03010-t001:** Patient, tumor, and treatment characteristics.

	Baseline		6 Weeks		2 Years	
Characteristic	No.	%	No.	%	No.	%
Patient Characteristics						
Age at diagnosis						
Median (years)	55.0		54.0		56.0	
Range (years)	27–81		30–78		30–78	
Tumor Characteristics						
Side						
Right breast	208	44.1	186	42.9	170	45.0
Left breast	264	55.9	247	57.1	208	55.0
Quadrant						
Lower inner	51	10.8	44	10.2	18	4.8
Lower outer	72	15.3	65	15.0	57	15.1
Upper inner	115	24.4	100	23.1	92	24.3
Upper outer	289	61.2	187	43.2	179	47.4
Multiple Quadrants	39	8.3	37	8.5	32	8.5
Pathologic stage						
≤pT1	347	73.5	320	73.9	278	73.5
pT2	117	24.8	106	24.5	95	25.2
pT3	5	1.1	4	0.9	3	0.8
pT4	3	0.6	3	0.7	2	0.5
pN0	377	79.9	343	79.2	302	79.9
pN1	76	16.1	72	16.6	60	15.9
pN2	16	3.4	15	3.5	13	3.4
pN3	3	0.6	3	0.7	3	0.8
Treatment Characteristics						
Axillary dissection						
Yes	101	21.4	97	22.4	79	20.9
No	371	78.6	336	77.6	299	79.1
Nodal irradiation						
Yes	67	14.2	62	14.3	55	14.6
No	405	85.8	371	85.7	323	85.4
Chemotherapy						
None	266	56.4	243	56.1	213	56.3
Neoadjuvant only	76	16.1	69	15.9	63	16.7
Adjuvant only	127	26.9	118	27.3	100	26.5
Neoadjuvant and adjuvant	3	0.6	3	0.7	2	0.5
Anthracycline-based chemotherapy						
Yes	135	28.6	118	27.3	105	27.8
No	337	71.4	315	72.7	273	72.2
Taxane-based chemotherapy						
Yes	159	33.7	139	32.1	132	34.9
No	313	66.3	294	67.9	246	65.1
Trastuzumab-based therapy						
Yes	25	5.3	21	4.8	15	4.0
No	447	94.7	412	95.2	363	96.0
Endocrine therapy						
Yes	330	69.9	298	68.8	265	70.1
No	142	30.1	135	31.2	113	29.9
Radiotherapy						
IMRT-SIB	235	49.8	213	49.2	192	50.8
3D-CRT-seqB	237	50.2	220	50.8	186	49.2
Total	472	100.0	433	100.0	378	100.0

Abbreviations: No.: number; IMRT-SIB: conventional fractionated intensity-modulated radiotherapy with simultaneously integrated boost; 3D-CRT-seqB: 3D-conformal radiotherapy with sequential boost; SIB: simultaneously integrated boost.

**Table 2 cancers-14-03010-t002:** Correlations between breast asymmetry indices and overall esthetic outcomes of BCCT.core, and between breast asymmetry indices and patients’ or physicians’ assessment scores, at the baseline, 6 weeks, and 2 years after radiotherapy (four-point scale: excellent–good–fair–poor).

	BCCT.Core		Patients’ Self-Assessment		Physicians’ Assessment	
	Pearson	*p*-Value	Pearson	*p*-Value	Pearson	*p*-Value
**Baseline (*n* = 472)**						
pBRA	0.447	**<0.001**	0.051	0.266	0.172	**<0.001**
LBC	0.669	**<0.001**	0.143	**0.002**	0.211	**<0.001**
UNR	0.535	**<0.001**	0.098	**0.032**	0.200	**<0.001**
BCE	0.239	**<0.001**	0.051	0.265	0.116	**0.012**
BCD	0.593	**<0.001**	0.070	0.126	0.201	**<0.001**
BAD	0.415	**<0.001**	0.026	0.579	0.125	**0.007**
BOD	0.489	**<0.001**	0.070	0.127	0.159	**0.001**
**6 Weeks after Radiotherapy (*n* = 433)**						
pBRA	0.321	**<0.001**	0.092	0.055	0.180	**<0.001**
LBC	0.554	**<0.001**	0.135	**0.005**	0.190	**<0.001**
UNR	0.391	**<0.001**	0.108	**0.025**	0.205	**<0.001**
BCE	0.134	**0.005**	0.117	**0.015**	0.196	**<0.001**
BCD	0.505	**<0.001**	0.082	0.088	0.141	**0.003**
BAD	0.394	**<0.001**	0.091	0.060	0.157	**0.001**
BOD	0.504	**<0.001**	0.108	**0.024**	0.157	**0.001**
**2 Years after Radiotherapy (*n* = 378)**						
pBRA	0.357	**<0.001**	0.203	**<0.001**	0.293	**<0.001**
LBC	0.688	**<0.001**	0.246	**<0.001**	0.283	**<0.001**
UNR	0.454	**<0.001**	0.201	**<0.001**	0.285	**<0.001**
BCE	0.147	**0.004**	0.154	**0.003**	0.219	**<0.001**
BCD	0.621	**<0.001**	0.301	**<0.001**	0.313	**<0.001**
BAD	0.507	**<0.001**	0.261	**<0.001**	0.284	**<0.001**
BOD	0.535	**<0.001**	0.246	**<0.001**	0.317	**<0.001**

A Pearson correlation score was used to determine the correlation between breast asymmetry indices and overall cosmetic outcomes of BCCT.core, as well as between breast asymmetry indices and patients’ or physicians’ assessment scores at the baseline, 6 weeks, and 2 years after radiotherapy. A two-sided significance test was used to compute the statistical significance. Statistically significant *p*-values (a ≤ 0.05) are presented in bold. Abbreviations: *n*: number of valid assessments; pBRA: breast retraction assessment; LBC: lower breast contour; UNR: upward nipple retraction; BCE: breast compliance evaluation; BCD: breast contour difference; BAD: breast area difference; BOD: breast overlap difference; Bold: The bold is for highlighting significant values.

**Table 3 cancers-14-03010-t003:** Correlations between breast asymmetry indices and overall esthetic outcomes of BCCT.core, and between breast asymmetry indices and patients’ or physicians’ assessment scores, at the baseline, 6 weeks, and 2 years after radiotherapy (two-point scale: excellent/good–fair/poor).

	BCCT.Core		Patients’ Self-Assessment		Physicians’ Assessment	
	Pearson	*p*-Value	Pearson	*p*-Value	Pearson	*p*-Value
**Baseline (*n* = 472)**						
pBRA	0.441	**<0.001**	−0.039	0.399	0.177	**<0.001**
LBC	0.590	**<0.001**	0.053	0.248	0.227	**<0.001**
UNR	0.524	**<0.001**	0.015	0.740	0.226	**<0.001**
BCE	0.243	**<0.001**	0.005	0.912	0.145	**0.002**
BCD	0.495	**<0.001**	0.004	0.937	0.214	**<0.001**
BAD	0.328	**<0.001**	−0.017	0.708	0.128	**0.005**
BOD	0.433	**<0.001**	0.010	0.820	0.169	**<0.001**
**6 Weeks after Radiotherapy (*n* = 433)**						
pBRA	0.321	**<0.001**	−0.007	0.890	0.152	**0.002**
LBC	0.573	**<0.001**	0.054	0.262	0.138	**0.004**
UNR	0.420	**<0.001**	0.032	0.504	0.209	**<0.001**
BCE	0.168	**<0.001**	0.082	0.090	0.230	**<0.001**
BCD	0.499	**<0.001**	0.026	0.588	0.128	**0.008**
BAD	0.405	**<0.001**	0.003	0.955	0.109	**0.024**
BOD	0.517	**<0.001**	0.030	0.536	0.040	0.403
**2 Years after Radiotherapy (*n* = 378)**						
pBRA	0.373	**<0.001**	0.176	**0.001**	0.260	**<0.001**
LBC	0.652	**<0.001**	0.232	**<0.001**	0.235	**<0.001**
UNR	0.457	**<0.001**	0.173	**0.001**	0.258	**<0.001**
BCE	0.210	**<0.001**	0.085	0.101	0.184	**<0.001**
BCD	0.568	**<0.001**	0.252	**<0.001**	0.245	**<0.001**
BAD	0.457	**<0.001**	0.220	**<0.001**	0.208	**<0.001**
BOD	0.530	**<0.001**	0.184	**<0.001**	0.238	**<0.001**

A Pearson correlation score was used to determine correlation between breast asymmetry indices and overall cosmetic outcomes of BCCT.core, as well as between breast asymmetry indices and patients’ or physicians’ assessment scores, at the baseline, 6 weeks, and 2 years after radiotherapy. A two-sided significance test was used to compute the statistical significance. Statistically significant *p*-values (a ≤ 0.05) are presented in bold. Abbreviations: N: number of valid assessments; pBRA: breast retraction assessment; LBC: lower breast contour; UNR: upward nipple retraction; BCE: breast compliance evaluation; BCD: breast contour difference; BAD: breast area difference; BOD: breast overlap difference; Bold: The bold is for highlighting significant values.

**Table 4 cancers-14-03010-t004:** Agreement of BCCT.core software results with patients’ or physicians’ assessment scores, and agreement of patients’ assessment scores with physicians’ assessment scores, at the baseline, 6 weeks, and 2 years after radiotherapy (four-point scale: excellent–good–fair–poor).

	a	k	*p*-Value (k)	wk	*p*-Value (wk)
**Baseline**					
BCCT.core vs. patients’ assessment	225/472	0.088	**0.019**	0.109	**0.003**
BCCT.core vs. physicians’ assessment	167/472	−0.029	0.178	0.027	0.173
patients’ assessment vs. physicians’ assessment	205/472	0.052	**0.024**	0.100	**<0.001**
**6 Weeks after Radiotherapy**					
BCCT.core vs. patients’ assessment	146/433	−0.002	0.952	0.024	0.321
BCCT.core vs. physicians’ assessment	157/433	0.051	0.060	0.084	**0.001**
patients’ assessment vs. physicians’ assessment	313/433	0.516	**<0.001**	0.572	**<0.001**
**2 Years after Radiotherapy**					
BCCT.core vs. patients’ assessment	149/378	0.052	0.138	0.111	**0.002**
BCCT.core vs. physicians’ assessment	151/378	0.069	**0.050**	0.138	**<0.001**
patients’ assessment vs. physicians’ assessment	274/378	0.543	**<0.001**	0.625	**<0.001**

Cohen’s Kappa statistic was used to determine agreement of the BCCT.core software results with patients’ or physicians’ assessment scores, as well as agreement of patients’ assessment scores with physicians’ assessment scores, at the baseline, 6 weeks, and 2 years after radiotherapy. Rates of total agreement, Kappa coefficients, and weighted Kappa coefficients with their respective *p*-values are indicated. Statistically significant *p*-values (a ≤ 0.05) are presented in bold. Abbreviations: a: rate of total agreement; k: Kappa coefficient; wk: weighted Kappa coefficient; *p*-value (k): *p*-value of the Kappa coefficient; *p*-value (wk): *p*-value of the weighted Kappa coefficient; vs.: versus; Bold: The bold is for highlighting significant values.

**Table 5 cancers-14-03010-t005:** Agreement of the BCCT.core software results with patients’ or physicians’ assessment scores, and agreement of patients’ assessment scores with physicians’ assessment scores, at the baseline, 6 weeks, and 2 years after radiotherapy (two-point scale: excellent/good–fair/poor).

	a	k	*p*-Value (k)	wk	*p*-Value (wk)
**Baseline**					
BCCT.core vs. patients’ assessment	397/472	−0.036	0.433	−0.036	0.433
BCCT.core vs. physicians’ assessment	393/472	0.144	**0.001**	0.144	**0.001**
patients’ assessment vs. physicians’ assessment	392/472	0.127	**0.004**	0.127	**0.004**
**6 Weeks after Radiotherapy**					
BCCT.core vs. patients’ assessment	304/433	−0.005	0.883	−0.005	0.883
BCCT.core vs. physicians’ assessment	309/433	0.072	0.080	0.072	0.080
patients’ assessment vs. physicians’ assessment	388/433	0.344	**<0.001**	0.344	**<0.001**
**2 Years after Radiotherapy**					
BCCT.core vs. patients’ assessment	301/378	0.213	**<0.001**	0.213	**<0.001**
BCCT.core vs. physicians’ assessment	301/378	0.244	**<0.001**	0.244	**<0.001**
patients’ assessment vs. physicians’ assessment	336/378	0.532	**<0.001**	0.532	**<0.001**

Cohen’s Kappa statistic was used to determine agreement of the BCCT.core software results with patients’ or physicians’ assessment scores, as well as agreement of patients’ assessment scores with physicians’ assessment scores, at the baseline, 6 weeks, and 2 years after radiotherapy. The rates of total agreement, Kappa coefficients, and weighted Kappa coefficients with their respective *p*-values are indicated. Statistically significant *p*-values (a ≤ 0.05) are presented in bold. Abbreviations: a: rate of total agreement; k: Kappa coefficient; wk: weighted Kappa coefficient; *p*-value (k): *p*-value of the Kappa coefficient; *p*-value (wk): *p*-value of the weighted Kappa coefficient; vs.: versus; Bold: The bold is for highlighting significant values.

## Data Availability

All data presented in this article are available on request to the editor or the reviewers. Open Access publishing is supported.

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
