# Peer review of "Methods of Esthetic Assessment after Adjuvant Whole-Breast Radiotherapy in Breast Cancer Patients: Evaluation of the BCCT.core Software and Patients’ and Physicians’ Assessment from the Randomized IMRT-MC2 Trial"

_cancers, 2022, doi:10.3390/cancers14123010_

Round 1

Reviewer 1 Report

The Authors investigated the accuracy in terms of cosmetic assessment of the BCCT.core software, compared to patient’d snf physician’s assessed ratings in breast cancer patients undergoing breast conserving surgery and post-operative radiotherapy, within the IMRT-MC2 trials. The article is of interest. Good manuscript. Few minor comments:

1)     Can you provide a brief description of dose/fractionation in the 2 treatment arms?

2)     Table 1. Please, add range to median value for age.

3)     I would remove dose/fractionation from Table.1 and rather put it in the textual part.

Reviewer 2 Report

The authors present a paper about "Methods of esthetic assessment after adjuvant whole breast radiotherapy in breast cancer patients: Evaluation of the BCCT.core software and patients’ and physicians’ assessment from the randomized IMRT-MC2 trial".

The topic is interesting because it is absolutely  important to empower the melting among objective methods, subjective methods and especially patients'perpective.

The sample size is adequate and the methods have been sufficently explanined.

I believe there could be a few points worth futher investigation as follows:

1) Which was the experience of the physicians who evaluated the patients (all radiation oncologists? also surgeons and/or medical oncologists? how many years of experience in the field?)

2) Was there any correlation between the breast size and the esthetic outcome?

3) Why did the authors choose to include both 3D conformal and IMRT treatment? they have different dosiemtric profiles which may result into different cometic outcomes: was there any difference between the two groups?

4) Was there any corelation with the skin photopype?

5) Was a psycho-oncologist involved in this study?

6) The authors state that "it may not sufficiently evaluate the esthetic outcome as perceived by patients. It may be more appropriate for a long-term follow-up, when symmetry seems to increase in importance". I believe that the use of a dedicated software should be considered an addition only in case it could anticipate something which may become clinically relevant in the future; if the software can only confirm what has already been perceieved by patients which is the usefulness of this software? Please clearly explain this point  

Round 2

Reviewer 2 Report

I have no further comments.